# Domain Transfer Learning for Hyperspectral Image Super-Resolution

**Xiaoyan Li [1,*], Lefei Zhang [2] and Jane You [3]**

1    School of Computer Science, China University of Geosciences, Wuhan 430074, China
2    School of Computer, Wuhan University, Wuhan 430072, China; zhanglefei@whu.edu.cn
3    Department of Computing, The Hong Kong Polytechnic University, Kowloon 999077, Hong Kong, China; csyjia@comp.polyu.edu.hk
*    Correspondence: lixy@cug.edu.cn; Tel.: +86-027-67883716

**Abstract:** A Hyperspectral Image (HSI) contains a great number of spectral bands for each pixel; however, the spatial resolution of HSI is low. Hyperspectral image super-resolution is effective to enhance the spatial resolution while preserving the high-spectral-resolution by software techniques. Recently, the existing methods have been presented to fuse HSI and Multispectral Images (MSI) by assuming that the MSI of the same scene is required with the observed HSI, which limits the super-resolution reconstruction quality. In this paper, a new framework based on domain transfer learning for HSI super-resolution is proposed to enhance the spatial resolution of HSI by learning the knowledge from the general purpose optical images (natural scene images) and exploiting the cross-correlation between the observed low-resolution HSI and high-resolution MSI. First, the relationship between low- and high-resolution images is learned by a single convolutional super-resolution network and then is transferred to HSI by the idea of transfer learning. Second, the obtained Pre-high-resolution HSI (pre-HSI), the observed low-resolution HSI, and high-resolution MSI are simultaneously considered to estimate the endmember matrix and the abundance code for learning the spectral characteristic. Experimental results on ground-based and remote sensing datasets demonstrate that the proposed method achieves comparable performance and outperforms the existing HSI super-resolution methods.

**Keywords:** Hyperspectral Image (HSI) super-resolution; domain transfer learning; convolutional super-resolution network; image fusion

## 1. Introduction

The hyperspectral imaging technique can acquire images with hundreds of spectral bands for each image pixel. Hyperspectral Images (HSI) have been widely used in numerous applications, such as urban planning, precision agriculture, and land-cover classification [1–6]. However, due to the limitations of the image spectrometer, it is difficult to acquire hyperspectral images with high-spectral-resolution and high-spatial-resolution simultaneously. Therefore, the spatial resolution of HSI data is often low, which is hard to capture the details of the land-cover and highly degrades the subsequent processing in the remote sensing fields. Hence, enhancing the spatial resolution of HSI data has gained more and more attention in the remote sensing community.

To achieve this goal, the HSI super-resolution technique [7,8] has been investigated to enhance the spatial resolution of HSI data by a software technique. The existing HSI super-resolution methods are based on the assumption that multiple observations of the same scene are required with the observed low-resolution HSI. These auxiliary observations, such as RGB images, panchromatic images (PAN), and Multispectral Images (MSI), can be used to estimate the missing spatial information in

the observed HSI data. A comparative review of the existing hyperspectral and multispectral image fusion approaches can be found in [9].

On the basis of spectral mixture analysis [10], the HSI data can be formulated by an endmember matrix multiplied by the corresponding abundance matrix. The endmember matrix indicates the pure spectral signatures, while the abundance matrix denotes the proportions of endmember spectra for each pixel. Therefore, the problem of HSI super-resolution is reduced to obtaining the optimal endmember and abundance matrices of the original high-spatial-resolution HSI.

Recently, many HSI approaches have been presented to enhance the spatial resolution of the observed HSI by exploiting the auxiliary observations. Since the observed HSI and the target HSI capture the same scene, their endmember matrices should be the same [11]. Moreover, the abundance matrix is often estimated from the observed MSI. Aiazzi et al. [12] presented an HSI super-resolution method based on a Generalized Laplacian Pyramid (GLP). Zhao et al. [13] proposed an HSI super-resolution approach by introducing a spatial-spectral joint nonlocal similarity in the reconstruction. Simões et al. [11] proposed a Hyperspectral Super-resolution (HySure) method with a convex subspace-based formulation. HySure introduced a total variation regularization for the estimation of the abundance matrix, which is valid to preserve edges while suppressing noise in the homogeneous regions. Generally, the subspace transformation is derived from the low-spatial-resolution HSI by an endmember extraction technique, i.e., Vertex Component Analysis (VCA) [14]. Lanaras et al. [15] proposed a Proximal Alternating Linearized Minimization (PALM) for HSI super-resolution, which jointly unmixes the observed HSI and MSI into the spectra of endmembers and the corresponding fractional abundances. A nonnegative structural sparse representation-based HSI super-resolution method was presented in [16]. Gao et al. [17] proposed a Self-Dictionary Sparse Regression (SDSR) method by combining the observed HSI and MSI to estimate the endmember matrix and the abundance matrix.

Compared with the auxiliary observations, general purpose optical images (natural scene images) are easily acquired and have more rich information. The relationship between low- and high-resolution HSIs is assumed to be the same as that between low- and high-resolution natural scene images [18]. That is because the optical information in remote sensing images has strong similarities with that in natural images [11]. In [18], the target high-spatial-resolution HSI was estimated by introducing a Collaborative Nonnegative Matrix Factorization (CNMF) between the observed HSI and the transferred high-spatial-resolution HSI, without requiring any auxiliary observations. However, it usually ignores the abundance information estimated from the observed MSI, which is effective to improve the final super-resolution reconstruction quality. To exploit the cross-correlation between the observed HSI and MSI and the nonlinear mapping relationship in the natural image domain effectively, a new framework based on domain transfer learning is proposed to enhance the spatial similarity and preserve the spectral consistency in the reconstructed HSIs. The main contributions of this paper can be summarized as follows:

(1) To reduce the blurring effect in the observed HSI, the proposed method first exploits a Convolutional Super-resolution Network (CSN) model trained by the natural scene images. Then, the trained CSN model is transferred to the HSI domain.

(2) The observed low-spatial-resolution HSI and the transferred pre-high-spatial-resolution HSI are simultaneously used to estimate the optimal endmember matrix with higher precision compared with the baselines.

(3) Considering the spatial information and spectral consistency, a new optimization function with two regularization parameters is proposed to obtain the optimal endmember and abundance matrices of the target HSI with higher spatial resolution, while preserving its high-spectral-resolution.

The remainder of this paper is organized as follows. Section 2 briefly introduces some related works. Section 3 provides our proposed method in detail. In Section 4, experimental results are presented to demonstrate the effectiveness of the proposed method compared with state-of-the-art baselines. Finally, a conclusion of this work is provided in Section 5.

## 2. Related Works

In the past decades, many image super-resolution methods have been presented. According to whether the auxiliary observations are required, the existing image super-resolution methods are divided into two categories: single-image-based and auxiliary-based. The single-image-based approaches (e.g., [19,20]) try to improve the spatial-resolution of each band image in HSI data. However, these methods ignore the spectral information in the reconstruction process.

The auxiliary-based HSI super-resolution methods have been proposed to estimate the high-spatial-resolution HSI by fusing the observed HSI and the high-spatial-resolution auxiliary observations, i.e., PAN [21], RGB, and MSI [22]. Thomas et al. [23] introduced a component substitution scheme for HSI super-resolution. The observed HSI is first divided into spatial and spectral components. Then, the estimated HSI is obtained by substituting the spatial component with the observed PAN. Based on the spectral mixture analysis, Yokoya et al. [24] proposed an HSI super-resolution method by constructing a coupled feature space between the observed HSI and MSI of the same scene. Subsequently, many HSI super-resolution approaches have been proposed to develop the efficient estimation of the endmember and abundance matrices with some constraints [11–13,15–18]. For example, in [11], a spatial smoothness constraint was imposed in the abundance optimization. In [25], the nonnegativity and sparsity constraints were introduced in a constrained sparse representation for HSI super-resolution. Fang et al. [26] proposed a superpixel-based sparse representation model to fuse the observed HSI and MSI. Yi et al. [27] presented a regularization model by exploiting the spatial and spectral correlations to achieve HSI-MSI fusion. These methods directly use the observed HSI or the bicubic interpolation of the observed HSI to estimate the sparse dictionary or the endmember matrix. However, a serious blurring effect in the observed HSI may lead to low estimation accuracy of the endmember matrix.

To sum up, it is important to capture the spatial similarity of each band image and the cross-correlation between the observed HSI and the auxiliary observations. Therefore, this is an effective way to take full use of the advantages of the single-image-based and auxiliary-based approaches.

## 3. Proposed Method

To take advantage of the additional information from the natural images and the spatial-spectral information between the observed HSI and MSI, a new *Super-Resolution* network based on *Domain Transfer Learning* (SRDTL) is proposed to enhance the spatial resolution, while preserving the abundant spectral information in the reconstructed hyperspectral remote sensing images. The overall flowchart of the proposed method is shown in Figure 1.

### 3.1. Notation

Let $\{X_s, Y_s\}$ be the high- and low-resolution information in the natural image domain. Similarly, $\{X, Y\}$ is denoted as the high- and low-resolution information in the HSI domain, where $X \in \mathbb{R}^{M \times N \times L}$ and $Y \in \mathbb{R}^{m \times n \times L}$ are the original high-spatial-resolution HSI and the observed low-spatial-resolution HSI, $M$, $N$, $m$, $n$ represent two spatial sizes, and $L$ denotes the number of spectral bands. $X$ and $Y$ have the same high-spectral-resolution. Based on these observations: $m \ll M$ and $n \ll N$, the super-resolution problem, i.e., the estimation of $X$, is severely ill-posed. For convenience, the HSI data are converted from 3D to 2D by concatenating the spatial pixels for each spectral band, i.e., $X \in \mathbb{R}^{L \times MN}$ and $Y \in \mathbb{R}^{L \times mn}$. In addition, $Z \in \mathbb{R}^{l \times MN}$ represents the observed MSI with high-spatial-resolution, but low-spectral-resolution, where $l$ is the number of spectral

bands in MSI. In particular, the observed HSI $Y$ and the observed MSI $Z$ are degraded from the original high-spatial-resolution and high-spectral-resolution HSI $X$ [11,24]. Therefore, the observed low-spatial-resolution HSI $Y$ and low-spectral-resolution MSI $Z$ can be represented as:

$$Y = \Psi(X), \qquad Z = \Phi(X), \tag{1}$$

where $\Psi : \mathbb{R}^{L \times MN} \to \mathbb{R}^{L \times mn}$ and $\Phi : \mathbb{R}^{L \times MN} \to \mathbb{R}^{l \times MN}$ are two mapping functions, which may be linear or nonlinear.

Generally, the MSI $Z$ is often approximated as:

$$Z \approx HX, \tag{2}$$

where $H \in \mathbb{R}^{l \times L}$ is the spectral response matrix, which is often assumed to be known [11,28]. This means MSI $Z$ can be easily obtained by the spectral degradation of the original high-spectral-resolution HSI $X$.

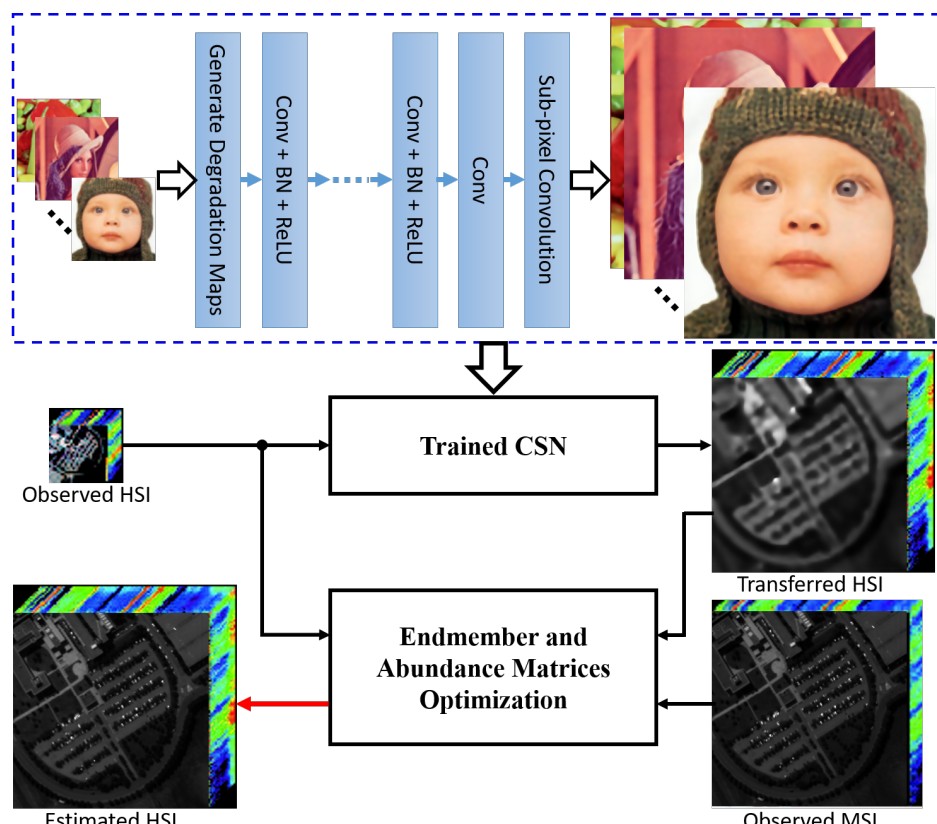

**Figure 1.** The flowchart of the proposed method. First, the Convolutional Super-Resolution Network (CSN) model parameters are learned on the low- and high-resolution natural images. Then, the transferred HSI is obtained by the trained CSN model with respect to the observed HSI. Finally, the endmember matrix and the abundance matrix of the estimated HSI are optimized by using the observed HSI, the observed Multispectral Image (MSI), and the transferred HSI. BN, Batch Normalization.

As is known in [11,17,28,29], the spectrum at each pixel position is often assumed to be a linear combination of several endmember spectra. Therefore, the high-spatial-resolution and high-spectral-resolution HSI $X$ is formulated as:

$$X \approx UV, \tag{3}$$

where $\boldsymbol{U} \in \mathbb{R}^{L \times C}$ is the endmember matrix, $\boldsymbol{V} \in \mathbb{R}^{C \times MN}$ is the abundance matrix, and $C$ represents the total number of endmembers. $\boldsymbol{U}$ denotes the spectral signatures of the underlying materials, while $\boldsymbol{V}$ represents the proportions of endmember spectra in each spatial point of the scene.

*3.2. Domain Transfer Learning*

Inspired by the idea of transfer learning [18,30], the mapping between low- and high-resolution images can be learned in the natural image domain and then transferred to the hyperspectral image domain. Moreover, deep learning has strong generalization ability and representation power to achieve domain transfer learning. Recently, many deep convolutional neural networks (CNNs) for image super-resolution have been presented, e.g., [20,31]. The effective CNNs can capture the nonlinear mapping between low- and high-resolution images.

To better learn the nonlinear mapping between low- and high-resolution images, a deep Convolutional Super-resolution Network (CSN) [20] is constructed to handle multiple and even spatially-variant degradations, which significantly enhances the applicability in the real world. In the test process, CSN takes a low-resolution natural image and the degradation maps as the input and then produces the corresponding high-resolution natural image. The degradation maps contain the warping knowledge, which can enable the super-resolution network to have the spatial transformation ability [32]. Similar to [20,33], a cascade of $3 \times 3$ convolutional layers are used to perform the nonlinear mapping between low- and high-resolution images. Each layer includes three operations: Convolution (Conv), Batch Normalization (BN) [34], and Rectified Linear Units (ReLU) [35]. Generally, "Conv + BN + ReLU" is applied to each convolutional layer excluding the last convolutional layer with only one "Conv" operation. Then, an additional sub-pixel convolutional layer is applied to convert several high-resolution subimages with a size of $m \times n \times r^2 K$ into a single high-resolution image with a size of $M \times N \times K$, where $K$ is the number of channels and $r$ denotes the magnification factor, which is equal to $M/m$ and $N/n$. Since CSN operates on RGB channels rather than the luminance channel (i.e., Y channel in YCbCr color space), the value of $K$ is three.

The CSN directly learns the nonlinear mapping between the low-resolution image $\boldsymbol{Y}_s \in \mathbb{R}^{m \times n \times 3}$ and the corresponding high-resolution image $\boldsymbol{X}_s \in \mathbb{R}^{M \times N \times 3}$. $\mathcal{F}(\cdot)$ denotes a deep CSN, which takes $\boldsymbol{Y}_s$ as the input and outputs the estimated high-resolution image. As is known to us, the nonlinear mapping function $\mathcal{F}(\cdot)$ can be learned by minimizing the loss function between the original high-resolution natural image $\boldsymbol{X}_s$ and the estimated high-resolution image $\mathcal{F}(\boldsymbol{Y}_s, \mathcal{M}; \Theta)$, where $\mathcal{M}$ represents the degradation maps with respect to the input image $\boldsymbol{Y}_s$ and $\Theta$ denotes the CSN model parameters. Given a large training set containing lots of natural images $\boldsymbol{D}_s = \left\{ \left( \boldsymbol{X}_s^i, \boldsymbol{Y}_s^i \right) | i = 1, 2, \cdots, N_s \right\}$, the model parameters $\Theta$ of a CSN model are estimated by solving the following minimization problem using the Adam method [36]:

$$\Theta = \arg \min_{\Theta} \frac{1}{2N_s} \sum_{i=1}^{N_s} \left\| \mathcal{F} \left( \boldsymbol{Y}_s^i, \mathcal{M}_i; \Theta \right) - \boldsymbol{X}_s^i \right\|_F^2, \tag{4}$$

where $\| \cdot \|_F$ denotes the Frobenius norm and $N_s$ is the number of training sample pairs.

Once having obtained the learned CSN model, i.e., the model parameters $\Theta$, it is reasonable to transfer it from the natural image domain to the HSI domain in each spectrum direction. In this paper, each band image of the low-spatial-resolution HSI $\boldsymbol{Y}$ is first copied to RGB channels as the input of the learned CSN model. Then, the estimated high-resolution band image is obtained by computing the mean value of the output RGB channels. In this way, the transferred high-spatial-resolution HSI $\boldsymbol{X}_h \in \mathbb{R}^{L \times MN}$ is predicted as:

$$\boldsymbol{X}_h = \left[ E \left( \mathcal{F} \left( \Gamma \left( \boldsymbol{Y}_1 \right), \mathcal{M}_1; \Theta \right) \right); E \left( \mathcal{F} \left( \Gamma \left( \boldsymbol{Y}_2 \right), \mathcal{M}_2; \Theta \right) \right); \cdots; E \left( \mathcal{F} \left( \Gamma \left( \boldsymbol{Y}_L \right), \mathcal{M}_L; \Theta \right) \right) \right], \tag{5}$$

where $\boldsymbol{Y}_i \in \mathbb{R}^{1 \times mn}$ means the $i^{\text{th}}$ band image of the low-spatial-resolution HSI $\boldsymbol{Y}$, $i = 1, 2, \cdots, L$. In addition, $\Gamma(\cdot)$ represents the composite function that first converts the vector from 1D to 2D, i.e.,

$\mathbb{R}^{1 \times mn} \to \mathbb{R}^{m \times n}$, and then copies the same matrix to RGB channels, while $E(\cdot)$ denotes the other composite function that first computes the mean value of the output RGB channels and then converts the matrix from 2D to 1D, i.e., $\mathbb{R}^{M \times N} \to \mathbb{R}^{1 \times MN}$.

### 3.3. Optimization

Since the low-spatial-resolution HSI $Y$ and the corresponding high-spatial-resolution HSI $X$, as well as the transferred high-spatial-resolution HSI $X_h$ capture the same scene, the underlying materials, i.e., the endmember matrix, should be the same. In addition, the abundance matrix of the transferred HSI $X_h$ is approximated as that from the estimated HSI $X$: $V_h \approx V$. Therefore, the low-spatial-resolution HSI $Y$ and the transferred HSI $X_h$ should be approximated as:

$$X_h \approx UV, \tag{6}$$

$$Y \approx UW, \tag{7}$$

where $W \in \mathbb{R}^{C \times mn}$ is the abundance matrix of the observed low-spatial-resolution HSI $Y$. Similarly, the observed MSI $Z$ and the desired high-spatial-resolution HSI $X$ share the same abundance matrix [17]. Hence, Equation (2) is changed as:

$$Z \approx U_m V, \tag{8}$$

where $U_m \in \mathbb{R}^{l \times C}$ is the endmember matrix of the observed MSI $Z$.

Combining Equations (6)–(8), the super-resolution problem for HSI can be solved by minimizing the following optimization problem:

$$\min_{U,V,W,U_m} \|X_h - UV\|_F^2 + \alpha \|Y - UW\|_F^2 + \beta \|Z - U_m V\|_F^2, \tag{9}$$
$$s.t. \quad U \geq 0, \ V \geq 0, \ W \geq 0, \ U_m \geq 0,$$

where $\alpha$ and $\beta$ are two nonnegative regularization parameters. The constraints $U \geq 0, V \geq 0, W \geq 0$, and $U_m \geq 0$ mean that these matrices are element-wise nonnegative [37]. In the linear mixing model, the abundance matrices $V$ and $W$ represent the abundances of the underlying materials that are necessarily nonnegative. Recent studies [29,38] have shown that multiplicative update rules are guaranteed to minimize the residual errors in Equation (9). Therefore, the multiplicative update rules for $U, V, W$, and $U_m$ are given as:

$$U \leftarrow U.* \left(\alpha YW^T + X_h V^T\right)./\left(\alpha UWW^T + UVV^T\right), \tag{10}$$

$$U_m \leftarrow U_m.* \left(ZV^T\right)./\left(U_m VV^T\right), \tag{11}$$

$$W \leftarrow W.* \left(U^T Y\right)./\left(U^T UW\right), \tag{12}$$

$$V \leftarrow V.* \left(U^T X_h + \beta U_m^T Z\right)./\left(U^T UV + \beta U_m^T U_m V\right), \tag{13}$$

where $(\cdot)^T$ denotes the function of transposition and $.*$ and $./$ represent the element-wise multiplication and division, respectively.

In summary, the overall description of the proposed method is given in Algorithm 1. Specifically, the matrices $U, V, W$, and $U_m$ are first initialized randomly and then updated by computing Equations (10)–(13) until convergence is reached. In this paper, the convergence condition is defined as the change ratio of the loss function being smaller than a given threshold $10^{-8}$. The optimal endmember matrix $\widehat{U}$ and abundance matrix $\widehat{V}$ are combined to reconstruct the high-spatial-resolution HSI $\widehat{X}$.

---

**Algorithm 1:** The proposed framework for HSI super-resolution.

---

**Input:** The observed low-spatial-resolution HSI $Y$; observed high-spatial-resolution MSI $Z$;
model parameters of CSN $\Theta$; regularization parameters $\alpha$ and $\beta$; number of
endmembers $C$; maximum number of iterations $\mathbb{T}$; convergence threshold $\tau$.

**Output:** Estimate the high-spatial-resolution HSI $\widehat{X}$.

**Transferring:**

    1. Obtain the transferred high-spatial-resolution HSI $X_h$ using Equation (5);

**Initialize:**

    2. Set $t = 0$ and $\varepsilon_0 = 0$;
    3. Initialize $U$, $U_m$, $W$, and $V$ randomly.

**for** $t = 1, 2, \cdots, \mathbb{T}$ **do**

    4. Update the endmember matrix $U$ of the estimated HSI using Equation (10);
    5. Update the endmember matrix $U_m$ of the observed MSI using Equation (11);
    6. Update the abundance matrix $W$ of the observed HSI using Equation (12);
    7. Update the abundance matrix $V$ of the estimated HSI using Equation (13);
    8. Compute the loss function $\varepsilon = \|Y - U * W\|_F^2 + \|Z - U_m * V\|_F^2$;
       **if** $t > 2$ & $(\varepsilon_0 - \varepsilon)/\varepsilon < \tau$ **then**
       break;

       **end**
    9. Set $\varepsilon_0 = \varepsilon$ and $t = t + 1$.

**end**

    10. Obtain the optimal matrices $\widehat{U} = U$ and $\widehat{V} = V$;
    11. Estimate the high-spatial-resolution HSI $X$ by computing $\widehat{X} = \widehat{U}\widehat{V}$.

---

## 4. Experiments

This section introduces the experiments for HSI super-resolution to verify the effectiveness of the proposed method. Three public HSI datasets are described in Section 4.1. Section 4.2 introduces state-of-the-art competing methods and evaluation indexes used in this paper. Section 4.3 shows the parameter analysis of the proposed method. Finally, Section 4.4 displays the experimental results.

### 4.1. HSI Datasets

In the experiments, three HSI datasets, i.e., CAVE (http://www.cs.columbia.edu/CAVE/databases/multispectral/), Pavia (http://www.ehu.eus/ccwintco/uploads/e/e3/Pavia.mat, and Paris (https://github.com/alfaiate/HySure), were used to evaluate the performance of the proposed method and state-of-the-art approaches. Commonly, the HSIs in these datasets were considered as the original high-spatial-resolution and high-spectral-resolution HSIs. The simulated low-spatial-resolution HSIs were obtained by first blurring the original HSIs and then downsampling the result with a magnification factor $r$ in each band direction. In addition, the value of HSI data was often large; therefore, the original HSIs should be normalized to $[0, 1]$. For better evaluation, all the output images were converted into 8-bit images.

**The CAVE dataset** [15,39] contains 32 HSIs with a size of $512 \times 512 \times 31$, which means that the size of each band image is $512 \times 512$ and the number of spectral bands is 31. Similar to [28], the high-spatial-resolution and low-spectral-resolution MSI was created by using the Nikon D700 spectral response (https://www.maxmax.com/spectral_response.htm), i.e., the spectral response

matrix $H$ in Equation (2). Therefore, the auxiliary images of the same scene used in the CAVE dataset are RGB images. To give better visual illustration, the Face image from the CAVE dataset is shown in Figure 2, where three spectral bands, i.e., 3, 25, and 30, were chosen to provide a pseudo-color view.

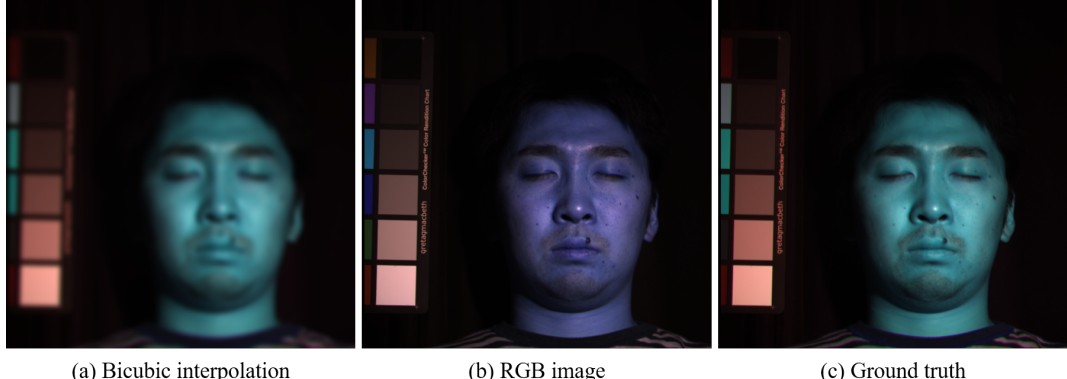

| (a) Bicubic interpolation | (b) RGB image | (c) Ground truth |

**Figure 2.** Face image from the CAVEdataset, where the color image consists of Spectral Bands 3, 25, and 30 for the red, green, and blue channels, respectively. (**a**) Bicubic interpolation of the observed low-spatial-resolution HSI, with a magnification factor of three; (**b**) RGB image after applying Nikon 700 spectral response; (**c**) the original high-spatial-resolution HSI considered as the ground truth.

**The Pavia** dataset [11,18] captures the urban area of the university of Pavia, which is commonly used in the application of hyperspectral classification. This dataset has a single HSI with a size of $610 \times 340 \times 103$; however, there are 10 non-informational spectral bands. A region of interest with an image size of $200 \times 200$ was selected from the original Pavia image, which contained valuable information. Therefore, a part of a size of $200 \times 200 \times 93$ was used as the referenced high-spatial-resolution HSI, after cropping the original Pavia image and removing the first ten spectral bands, i.e., $X = X(411{:}610, 141{:}340, 11{:}103)$. **The Paris** dataset [11] is often used for real data experiments. The size of the Paris image is $72 \times 72 \times 128$. For the two remote sensing datasets, i.e., Pavia and Paris, the MSIs were created by using the spectral response of the IKONOS satellite [11]. These MSIs contain four bands: blue, green, red, and near-infrared components. Figures 3 and 4 show the Pavia image and the Paris image, respectively. For the Pavia HSI data, three spectral bands 20, 16, and 5 were selected as a pseudo-color view. For Paris HSI data, the color image is composed of Bands 28, 13, and 3 for red, green, and blue channels, respectively.

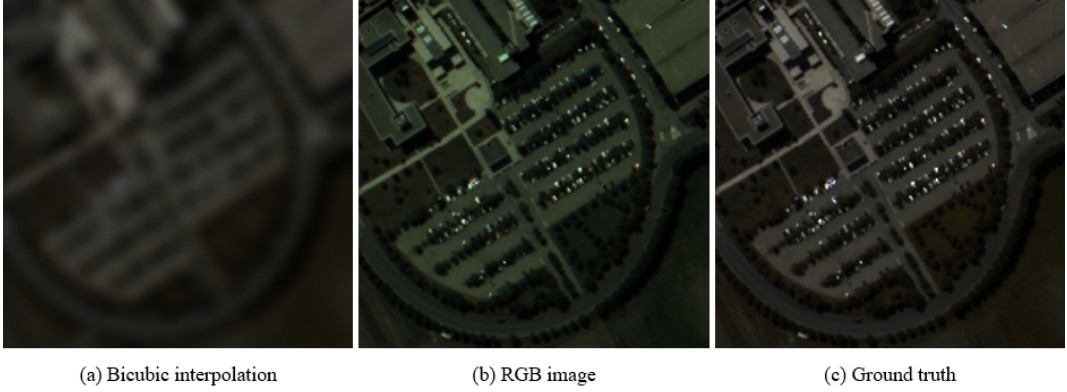

| (a) Bicubic interpolation | (b) RGB image | (c) Ground truth |

**Figure 3.** Pavia image, where the color image consists of Spectral Bands 20, 16, and 5 for the red, green, and blue channels, respectively. (**a**) Bicubic interpolation of the observed low-spatial-resolution HSI with a magnification factor of three; (**b**) MSI after applying IKONOS spectral response; only red, green, and blue components displayed; (**c**) the original high-spatial-resolution HSI considered as the ground truth.

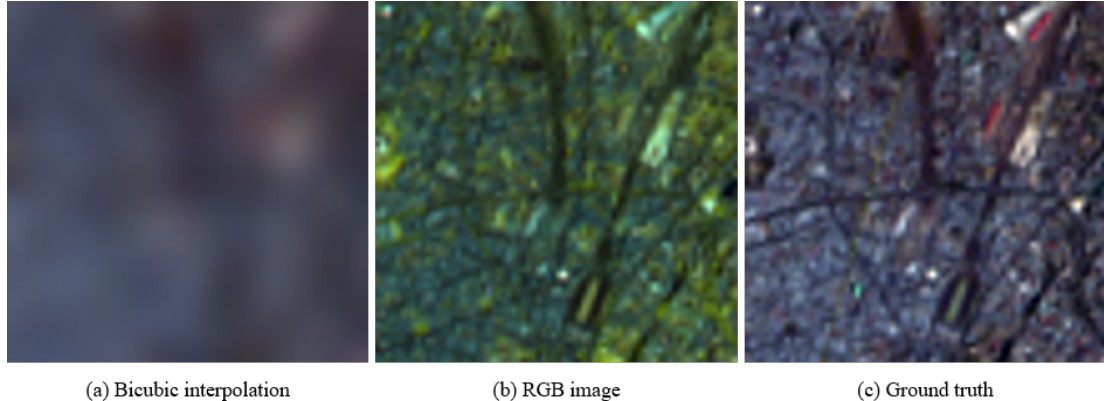

(a) Bicubic interpolation      (b) RGB image      (c) Ground truth

**Figure 4.** Paris image, where the color image consists of Spectral Bands 28, 13, and 3 for the red, green, and blue channels, respectively. (**a**) Bicubic interpolation of the observed low-spatial-resolution HSI with a magnification factor of three; (**b**) MSI after applying IKONOS spectral response; only red, green, and blue components displayed; (**c**) the original high-spatial-resolution HSI considered as the ground truth.

### 4.2. Competitors and Evaluation Indexes

**Competitors:** To evaluate the effectiveness of the proposed method, seven current state-of-the-art super-resolution approaches, including the conventional bicubic interpolation, Generalized Laplacian Pyramid (GLP) [12], Hyperspectral Super-resolution (HySure) [11], Proximal Alternating Linearized Minimization (PALM) [15], Collaborative Nonnegative Matrix Factorization (CNMF) [18], Self-Dictionary Sparse Regression (SDSR) [17], and Super-Resolution network for Multiple Noise-Free Degradation (SRMDNF) [20], were compared.

All of the above competing methods were obtained from the authors' MATLAB+MEX implementation, where the parameters were set for the best performance, as described in the corresponding references. For a pair comparison, the number of endmembers $C$ was set to 10 in this paper. In addition, all the experiments were performed on a personal computer with NVIDIA GeForce GTX 1080 Ti GPU, 64 GB memory, and 64-bit Windows 7 using MATLAB R2017b.

**Evaluation indexes:** There are many indexes to evaluate the image quality of the reconstructed images, e.g., qualitative [40,41] and quantitative [42–45]. In this paper, six quantitative indexes were employed to evaluate the quality of the reconstructed HSIs.

The first two quantitative indexes were the Mean Root-Mean-Squared Error (MRMSE) and the Mean Peak Signal-to-Noise Ratio (MPSNR). The RMSE and PSNR between the original high-spatial-resolution HSI $\boldsymbol{X} \in \mathbb{R}^{L \times MN}$ and the high-spatial-resolution estimated HSI $\widehat{\boldsymbol{X}} \in \mathbb{R}^{L \times MN}$ are defined as:

$$\text{RMSE}(\boldsymbol{X}_i, \widehat{\boldsymbol{X}}_i) = \sqrt{\frac{\|\boldsymbol{X}_i - \widehat{\boldsymbol{X}}_i\|_2^2}{MN}}, \tag{14}$$

$$\text{MRMSE}(\boldsymbol{X}, \widehat{\boldsymbol{X}}) = \frac{1}{L} \sum_{i=1}^{L} \text{RMSE}(\boldsymbol{X}_i, \widehat{\boldsymbol{X}}_i), \tag{15}$$

$$\text{MPSNR}(\boldsymbol{X}, \widehat{\boldsymbol{X}}) = \frac{1}{L} \sum_{i=1}^{L} 20 * \log_{10} \left( \frac{\max(\boldsymbol{X}_i)}{\text{RMSE}(\boldsymbol{X}_i, \widehat{\boldsymbol{X}}_i)} \right), \tag{16}$$

where $\max(\cdot)$ is the maximum value of the input vector. MRMSE and MPSNR are based on mean squared error. A smaller MRMSE value and a larger MPSNR value indicate that the reconstruction quality of the estimated HSI $\widehat{\boldsymbol{X}}$ is better.

The third quantitative index is the Mean Structure Similarity (MSSIM), which is to measure the structural consistency between the estimated image and the reference image. The MSSIM index of HSI data is formulated as:

$$\text{MSSIM}(\boldsymbol{X}, \widehat{\boldsymbol{X}}) = \frac{1}{L} \sum_{i=1}^{L} \text{SSIM}\left(\varphi(\boldsymbol{X}_i), \varphi(\widehat{\boldsymbol{X}}_i)\right), \tag{17}$$

where $\boldsymbol{X}_i \in \mathbb{R}^{1 \times MN}$ and $\widehat{\boldsymbol{X}}_i \in \mathbb{R}^{1 \times MN}$ denote the $i^{\text{th}}$ band images of the original HSI and the estimated HSI, respectively. $\varphi(\cdot)$ is the function to convert a vector from $\mathbb{R}^{1 \times MN}$ to $\mathbb{R}^{M \times N}$. The details of the function $\text{SSIM}(\cdot, \cdot)$ were described in [42]. The value of MSSIM belongs to $[0, 1]$. A larger MSSIM value indicates that the estimated image is more similar to the reference image in the structure.

The fourth index is the relative dimensionless global error in synthesis (ERGAS) [43], defined as:

$$\text{ERGAS}(\boldsymbol{X}, \widehat{\boldsymbol{X}}) = \frac{100}{r} \sqrt{\frac{1}{L} \sum_{i=1}^{L} \frac{\text{RMSE}^2(\boldsymbol{X}_i, \widehat{\boldsymbol{X}}_i)}{\mu_{2,i}^2}}, \tag{18}$$

where $r$ is the magnification factor and $\mu_{2,i}$ denotes the mean value $\widehat{\boldsymbol{X}}_i$. ERGAS calculates the band-wise normalized RMSE and then divides it by the spatial factor between the high- and low-resolution HSIs.

The fifth quantitative index is the Universal Image Quality Index (UIQI) [44]. UIQI is performed on a sliding window with a size of $32 \times 32$ and averages on all the window positions and over all the spectral bands. Denote $\boldsymbol{X}_{i,j}$ and $\widehat{\boldsymbol{X}}_{i,j}$ as the $j^{\text{th}}$ window region of the $i^{\text{th}}$ band image in the original and estimated HSIs, respectively. Therefore, the equation of UIQI is given by:

$$\text{UIQI}(\boldsymbol{X}, \widehat{\boldsymbol{X}}) = \frac{1}{LP} \sum_{i=1}^{L} \sum_{j=1}^{P} \frac{4\rho_{ij}^2}{\sigma_{1,ij}^2 + \sigma_{2,ij}^2} \frac{\mu_{1,ij}\mu_{2,ij}}{\mu_{1,ij}^2 + \mu_{2,ij}^2}, \tag{19}$$

where $P$ is the number of window positions and $\rho_{ij}$ represents the covariance between the subimages $\boldsymbol{X}_{i,j}$ and $\widehat{\boldsymbol{X}}_{i,j}$. In addition, $\mu_{1,ij}$ and $\sigma_{1,ij}$ are the mean value and standard deviation of the reference band image $\boldsymbol{X}_{i,j}$, respectively. Meanwhile, $\mu_{2,ij}$ and $\sigma_{2,ij}$ are the mean value and standard deviation of the estimated band image $\widehat{\boldsymbol{X}}_{i,j}$, respectively.

The sixth quantitative index is the Spectral Angle Mapper (SAM) [45], which is commonly used to evaluate the spectral information preservation at each pixel. SAM measures the spectral similarity by calculating the angle between the reference pixel $\boldsymbol{X}^k \in \mathbb{R}^{L \times 1}$ and the estimated pixel $\widehat{\boldsymbol{X}}^k \in \mathbb{R}^{L \times 1}$, defined as:

$$\text{SAM}(\boldsymbol{X}, \widehat{\boldsymbol{X}}) = \frac{1}{MN} \sum_{k=1}^{MN} \arccos\left(\frac{(\boldsymbol{X}^k)^T \widehat{\boldsymbol{X}}^k}{\|\boldsymbol{X}^k\|_2 \|\widehat{\boldsymbol{X}}^k\|_2}\right). \tag{20}$$

The SAM value is given in degrees. When the SAM value is close to zero, this indicates high spectral quality in the estimated HSI.

In summary, when MPSNR, MSSIM, and UIQI values are larger, the estimated HSI is more similar to the reference HSI. Furthermore, the smaller MRMSE, ERGAS, and SAM values are, the better the reconstruction quality of the estimated HSI is.

*4.3. Parameter Analysis*

As shown in Algorithm 1, the two main parameters of the proposed method are the regularization parameters $\alpha$ and $\beta$. The value of $\alpha$ affects the estimation of endmember matrix, while the value of $\beta$ affects the contribution of the abundance matrix from the observed MSI. The value of regularization parameter $\alpha$ was tuned with the set $\{0.0001, 0.0002, 0.0005, 0.001, 0.005, 0.01, 0.05, 0.1, 1\}$. Meanwhile, the parameter $\beta$ was varied with the set $\{1, 10, 100, 500, 1000, 1500, 2000, 3000, 4000, 5000, 10,000\}$. Figure 5

shows the MRMSEs and MSSIMs with respect to the values of regularization parameters $\alpha$ and $\beta$ on the CAVE, Pavia, and Paris datasets, respectively.

From Figure 5, we can see that when the value of $\beta$ increased, the MRMSEs and MSSIMs displayed a small change with a fixed $\alpha$. This is because the abundance matrix estimated from the observed MSI generated a similar contribution in the final result. For the CAVE dataset, a peak value was generated in the curved surface at $\alpha = 10^{-4}$ and $\beta = 1500$. In addition, the curved surfaces of the Pavia and Paris datasets were similar. With a fixed $\alpha$ value, the MSSIMs began to increase slowly when the value of $\beta$ continued to increase. If the value of $\beta$ is large, the ability of the estimation of the abundance matrix from the observed MSI will be limited. Therefore, the regularization parameters $\alpha$ and $\beta$ were set to $10^{-4}$ and $10^{4}$ for both the Pavia and Paris datasets in the experiments.

*4.4. Experimental Results*

The experimental results on the three HSI datasets are summarized in this subsection. Tables 1–3 show the quantitative results (using Equations (15)–(20)) and the computation time of the proposed method and the baselines on the CAVE, Pavia, and Paris HSI datasets with a magnification factor of three, respectively. The reported evaluation values were calculated in the 8-bit resulting images. Although the bicubic, GLP, CNMF, and SDSR methods take less time, the reconstruction quality of these methods is very limited. Compared with the competing methods, the computation time of the proposed method was the largest. That is because the proposed method relies on the construction of the transferred HSI, which is obtained by the existing SRMDNF method for each spectral band. For the CAVE dataset, the values shown in Table 1 are the average values of the six quantitative indexes over the 32 hyperspectral images. Compared with state-of-the-art approaches, the proposed method achieved better performance for most cases, which indicates that the information transferred from the natural image domain can enhance the ability of HSI super-resolution. Especially, the proposed method obtained the best MSSIM index values in most of the cases, which demonstrates that the estimated HSIs obtained by the proposed method have better structural similarity and higher spatial-resolution.

**Table 1.** Quantitative results on the CAVE dataset with a magnification factor of 3. GLP, Generalized Laplacian Pyramid; HySure, Hyperspectral Super-resolution; PALM, Proximal Alternating Linearized Minimization; CNMF, Collaborative Nonnegative Matrix Factorization; SDSR, Self-Dictionary Sparse Regression; SRMDNF, Super-Resolution network for Multiple Noise-Free Degradation; MPSNR, Mean Peak Signal-to-Noise Ratio; UIQI, Universal Image Quality Index; SAM, Spectral Angle Mapper.

| Indexes | Bicubic | GLP | HySure | PALM | CNMF | SDSR | SRMDNF | Proposed |
|---------|---------|-----|--------|------|------|------|--------|----------|
| MRMSE | 7.02 | 5.01 | 6.50 | **2.36** | 2.44 | 6.06 | 4.32 | 2.75 |
| MPSNR | 31.10 | 33.84 | 31.38 | 37.70 | 37.63 | 32.04 | 35.26 | **38.00** |
| MSSIM | 0.9137 | 0.9517 | 0.9286 | 0.9543 | 0.9352 | 0.9206 | 0.9165 | **0.9577** |
| ERGAS | 9.71 | 6.95 | 8.99 | **4.59** | 4.71 | 9.57 | 6.82 | 5.05 |
| UIQI | 0.7221 | **0.8461** | 0.6935 | 0.7981 | 0.7320 | 0.6692 | 0.7114 | 0.7921 |
| SAM | 0.0863 | 0.1379 | 0.0941 | 0.1230 | 0.1893 | 0.4462 | 0.1227 | **0.0763** |
| Time | 0.2 | 6.3 | 572.4 | 208.8 | 68.1 | 5.4 | 1541.0 | 1596.3 |

**Table 2.** Quantitative results on the Pavia dataset with a magnification factor of 3.

| Indexes | Bicubic | GLP | HySure | PALM | CNMF | SDSR | SRMDNF | Proposed |
|---------|---------|-----|--------|------|------|------|--------|----------|
| MRMSE | 17.14 | 11.93 | 9.60 | 10.57 | 9.23 | 9.52 | 13.99 | **8.11** |
| MPSNR | 23.54 | 26.73 | **30.31** | 27.72 | 28.91 | 28.58 | 25.28 | 29.97 |
| MSSIM | 0.5293 | 0.7968 | 0.8177 | 0.8720 | 0.8910 | 0.9008 | 0.6574 | **0.9305** |
| ERGAS | 13.53 | 9.27 | 7.24 | 8.40 | 7.15 | 7.94 | 11.52 | **7.03** |
| UIQI | 0.6096 | 0.8214 | 0.8664 | 0.8664 | 0.8977 | 0.8869 | 0.7775 | **0.9304** |
| SAM | 0.1390 | 0.1066 | 0.1397 | 0.0976 | **0.0830** | 0.1376 | 0.1069 | 0.0985 |
| Time | 0.2 | 5.4 | 137.0 | 159.7 | 54.0 | 3.5 | 1985.9 | 1990.3 |

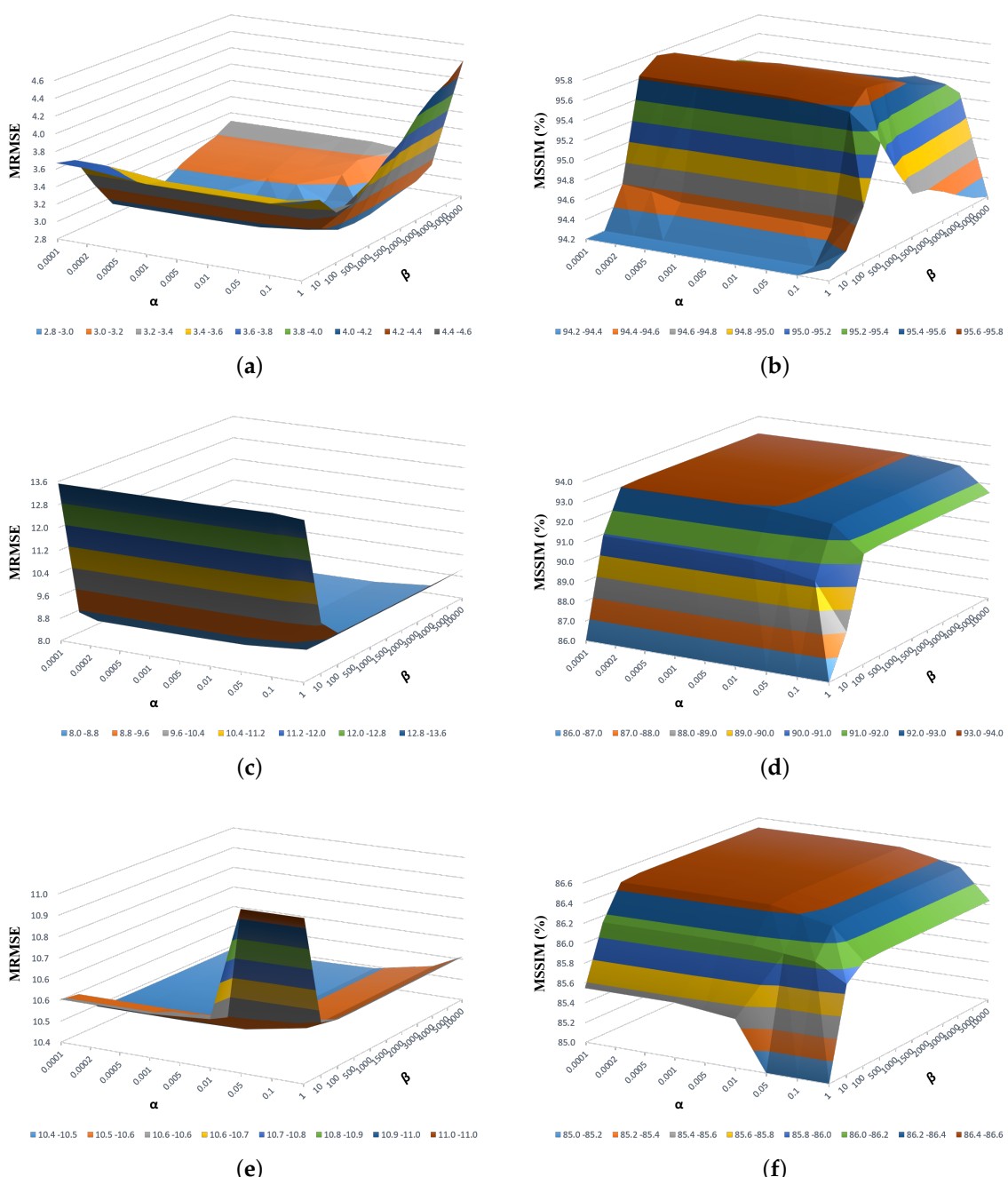

**Figure 5.** Mean Root-Mean-Squared Errors (MRMSEs) and Mean Structure Similarities (MSSIMs) versus the value of parameters $\alpha$ and $\beta$ using the proposed method with a magnification factor of three. (**a**) MRMSEs versus $\alpha$ and $\beta$ on the CAVE dataset. (**b**) MSSIMs (in terms of percentage) versus $\alpha$ and $\beta$ on the CAVE dataset. (**c**) MRMSEs versus $\alpha$ and $\beta$ on the Pavia dataset. (**d**) MSSIMs (%) versus $\alpha$ and $\beta$ on the Pavia dataset. (**e**) MRMSEs versus $\alpha$ and $\beta$ on the Paris dataset. (**f**) MSSIMs (%) versus $\alpha$ and $\beta$ on the Paris dataset.

**Table 3.** Quantitative results on the Paris dataset with a magnification factor of 3.

| Indexes | Bicubic | GLP | HySure | PALM | CNMF | SDSR | SRMDNF | Proposed |
|---------|---------|-----|--------|------|------|------|--------|----------|
| MRMSE | 19.05 | 13.38 | 13.74 | 13.61 | 10.64 | 14.53 | 17.31 | **10.47** |
| MPSNR | 22.81 | 25.82 | 26.39 | 25.61 | **27.97** | 25.42 | 23.60 | **27.97** |
| MSSIM | 0.3103 | 0.7305 | 0.6610 | 0.7642 | 0.8392 | 0.7454 | 0.3802 | **0.8648** |
| ERGAS | 9.74 | 6.90 | 7.56 | 7.12 | **5.53** | 7.67 | 8.95 | 5.62 |
| UIQI | 0.3129 | 0.6962 | 0.7386 | 0.6868 | 0.8314 | 0.6986 | 0.4699 | **0.8369** |
| SAM | 0.1248 | **0.0880** | 0.1173 | 0.0943 | 0.1004 | 0.1737 | 0.1150 | 0.1027 |
| Time | 0.2 | 2.0 | 6.8 | 15.5 | 2.0 | 0.5 | 251.3 | 255.4 |

For visualization, Figures 6–8 show the pseudo-color images obtained by different methods on the Balloons, Pavia, and Paris images with a magnification factor of three, respectively. In addition, Table 4 shows the MSSIM values corresponding to Figures 6–8. Although bicubic enlarges the image size, blurring effects often exist in the reconstructed images. GLP was better than bicubic; however, GLP easily produces the ghost effects on the edge area. HySure can learn high-spatial-resolution information from the observed MSI, but it may introduce noise in the final result. PALM performed worse on the structural regions than smooth regions. CNMF only uses the knowledge learned from natural images, which lacks some complex structural information. SDSR uses the result obtained by the bicubic interpolation as the pre-HSI, which will reduce the estimation accuracy of the endmember matrix. SRMD is a single-image super-resolution method, which learns more valuable information from a high-resolution training image dataset. However, it ignores the spectral consistency for HSI super-resolution. From the theory of domain transfer learning, the knowledge transferred from the natural image domain can improve the ability of HSI super-resolution reconstruction to a certain extent. For this reason, the proposed method combines the transferred HSI, the observed HSI, and MSI to estimate the optimal endmember and abundance matrices. Compared with state-of-the-art approaches, the proposed method achieves better performance for HSI super-resolution.

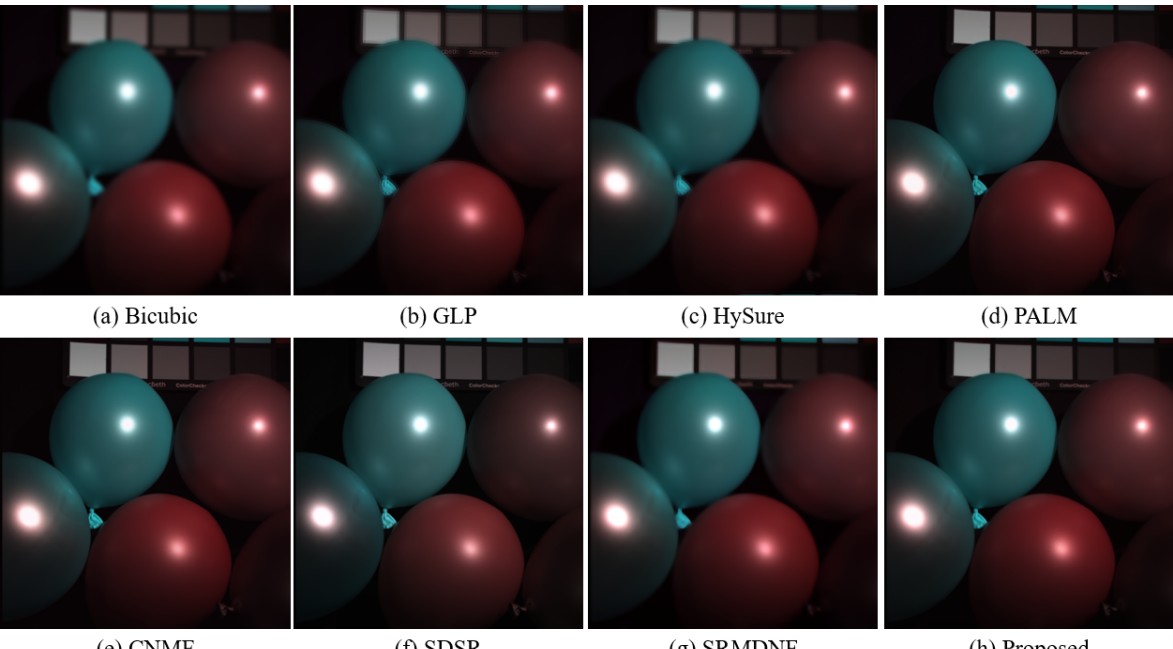

(a) Bicubic          (b) GLP          (c) HySure          (d) PALM

(e) CNMF          (f) SDSR          (g) SRMDNF          (h) Proposed

**Figure 6.** Super-resolution reconstruction results on the Balloons image from the CAVE dataset with a magnification factor of three. (**a**) Bicubic; (**b**) GLP; (**c**) HySure; (**d**) PALM; (**e**) CNMF; (**f**) SDSR; (**g**) SRMDNF; (**h**) proposed.

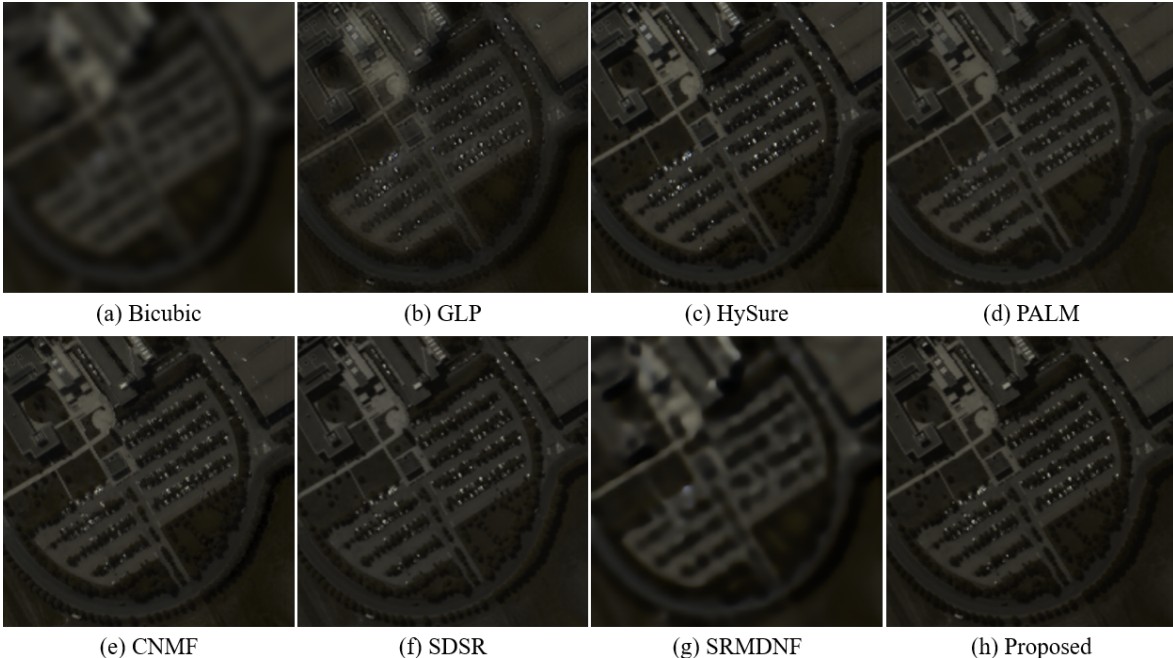

**Figure 7.** Super-resolution reconstruction results on the Pavia dataset with a magnification factor of three. (**a**) Bicubic; (**b**) GLP; (**c**) HySure; (**d**) PALM; (**e**) CNMF; (**f**) SDSR; (**g**) SRMDNF; (**h**) proposed.

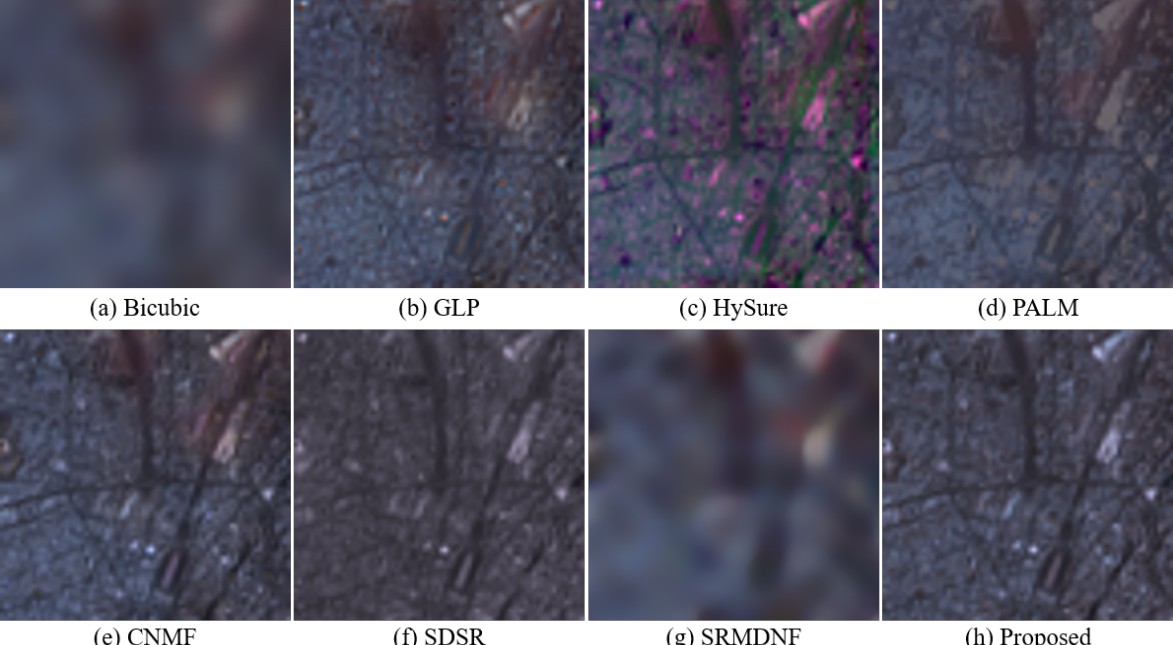

**Figure 8.** Super-resolution reconstruction results on the Paris dataset with a magnification factor of three. (**a**) Bicubic; (**b**) GLP; (**c**) HySure; (**d**) PALM; (**e**) CNMF; (**f**) SDSR; (**g**) SRMDNF; (**h**) proposed.

To further compare the performance of different methods in each spectral band, Figure 9 shows the RMSE between the original spectra and the estimated spectra on the Balloons, Face, Pavia, and Paris hyperspectral images. In addition, Table 5 shows the average RMSE values corresponding to Figure 9. Compared with the baselines, the proposed method preserved the spectral consistency from the observed HSI and achieved better performance for most cases.

In addition, to validate the effectiveness of the proposed method for different magnification factors, we repeated the experiments for a magnification factor of four. Tables 6–8 show the quantitative

results and the computation time of different methods on the CAVE, Pavia, and Paris datasets with a magnification factor of four, respectively. Further, Figure 10 shows the pseudo-color images obtained by different methods for the Face image from the CAVE dataset with a magnification factor of four. From the overall experimental results, we can see that the proposed method achieved better spatial similarity to the ground truth and preserved higher spectral consistency, which is suitable for the complex real-world HSI super-resolution tasks.

**Table 4.** The MSSIM values obtained by different methods on the Balloons, Pavia, and Paris images with a magnification factor of 3.

| Images | Bicubic | GLP | HySure | PALM | CNMF | SDSR | SRMDNF | Proposed |
|---|---|---|---|---|---|---|---|---|
| Balloons | 0.9612 | 0.9762 | 0.9632 | 0.9823 | 0.9771 | 0.9500 | 0.9678 | **0.9894** |
| Pavia | 0.5293 | 0.7968 | 0.8177 | 0.8720 | 0.8910 | 0.9008 | 0.6574 | **0.9305** |
| Paris | 0.3103 | 0.7305 | 0.6610 | 0.7642 | 0.8392 | 0.7454 | 0.3802 | **0.8648** |

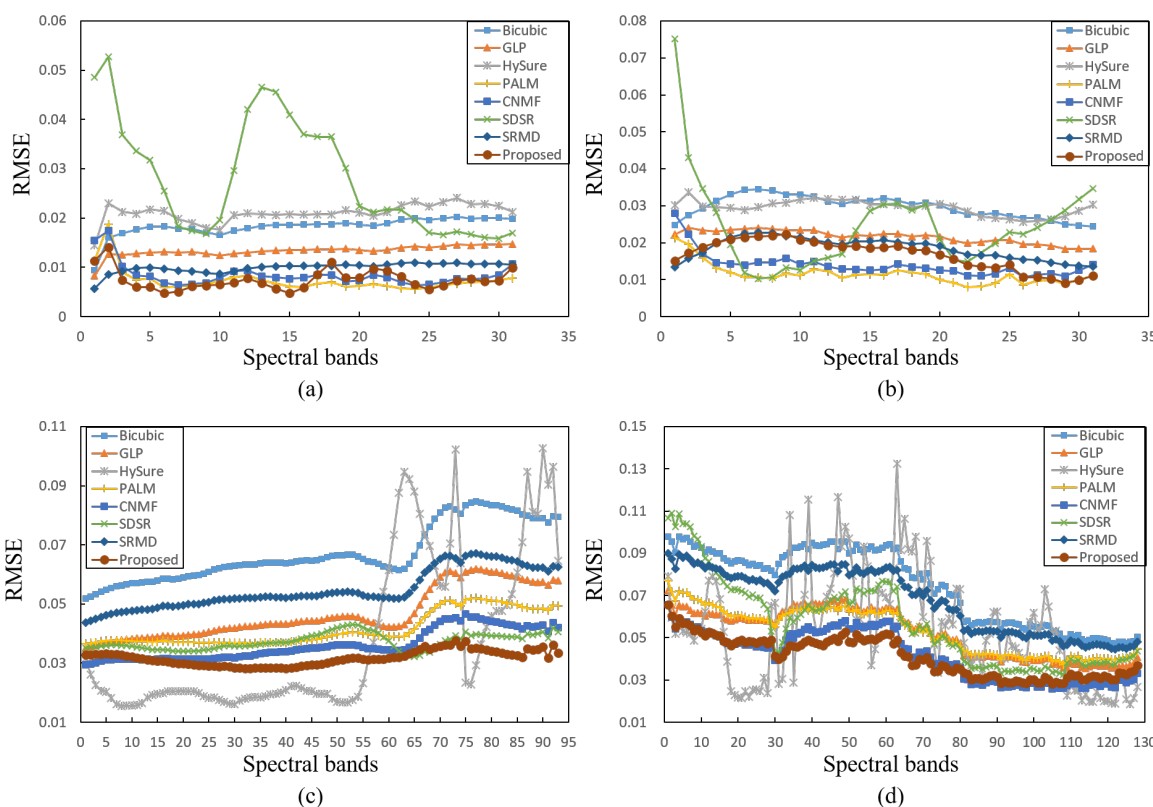

**Figure 9.** Spectral errors in terms of the RMSE for the compared methods. (**a**) Balloons image; (**b**) Face image; (**c**) Pavia image; (**d**) Paris image.

**Table 5.** The average values and the standard deviations of the RMSEs corresponding to Figure 9, in terms of percentage %.

| Images | Bicubic | GLP | HySure | PALM | CNMF | SDSR | SRMDNF | Proposed |
|---|---|---|---|---|---|---|---|---|
| Balloons | 1.83 ± 0.20 | 1.34 ± 0.12 | 2.10 ± 0.19 | 0.73 ± 0.24 | 0.84 ± 0.24 | 2.80 ± 1.15 | 0.99 ± 0.10 | **0.72 ± 0.21** |
| Face | 2.96 ± 0.31 | 2.16 ± 0.18 | 2.95 ± 0.20 | **1.15 ± 0.29** | 1.40 ± 0.34 | 2.44 ± 1.25 | 1.84 ± 0.30 | 1.67 ± 0.41 |
| Pavia | 6.72 ± 0.96 | 4.67 ± 0.84 | 3.76 ± 2.65 | 4.14 ± 0.56 | 3.61 ± 0.51 | 3.73 ± 0.27 | 5.48 ± 0.66 | **3.18 ± 0.25** |
| Paris | 7.47 ± 1.81 | 5.25 ± 1.16 | 5.38 ± 2.52 | 5.33 ± 1.09 | 4.17 ± 1.19 | 5.70 ± 2.05 | 6.79 ± 1.54 | **4.10 ± 0.94** |

**Table 6.** Quantitative results on the CAVE dataset with a magnification factor of 4.

| Indexes | Bicubic | GLP | HySure | PALM | CNMF | SDSR | SRMDNF | Proposed |
|---|---|---|---|---|---|---|---|---|
| MRMSE | 7.72 | 4.38 | 6.35 | 3.03 | 3.26 | 7.16 | 3.41 | **2.92** |
| MPSNR | 29.71 | 35.40 | 31.80 | 37.22 | 38.38 | 31.51 | 37.77 | **39.37** |
| MSSIM | 0.9112 | 0.9649 | 0.9390 | 0.9652 | 0.9417 | 0.9214 | 0.9667 | **0.9668** |
| ERGAS | 7.45 | 4.11 | 5.95 | 3.14 | 3.09 | 6.95 | 3.51 | **3.03** |
| UIQI | 0.6738 | **0.8645** | 0.6801 | 0.8208 | 0.7475 | 0.7026 | 0.7400 | 0.8187 |
| SAM | 0.0872 | 0.0742 | 0.0923 | 0.1086 | 0.1876 | 0.4478 | **0.0694** | 0.1564 |
| Time | 0.2 | 17.5 | 1038.3 | 452.5 | 137.6 | 12.3 | 1586.5 | 1722.2 |

**Table 7.** Quantitative results on the Pavia dataset with a magnification factor of 4.

| Indexes | Bicubic | GLP | HySure | PALM | CNMF | SDSR | SRMDNF | Proposed |
|---|---|---|---|---|---|---|---|---|
| MRMSE | 17.21 | 9.01 | 11.18 | 8.85 | 6.90 | 10.05 | 12.94 | **6.50** |
| MPSNR | 16.52 | 27.68 | 28.54 | 21.67 | 29.57 | 25.37 | 20.84 | **30.16** |
| MSSIM | 0.5252 | 0.8904 | 0.7870 | 0.9058 | 0.9427 | 0.8947 | 0.7092 | **0.9499** |
| ERGAS | 10.21 | 5.24 | 6.57 | 5.39 | 4.28 | 6.30 | 7.70 | **3.81** |
| UIQI | 0.5704 | 0.8968 | 0.7638 | 0.8968 | 0.9451 | 0.8554 | 0.8151 | **0.9531** |
| SAM | 0.1396 | 0.0872 | 0.1827 | 0.0877 | 0.0894 | 0.1479 | 0.1017 | **0.0649** |
| Time | 0.2 | 7.3 | 133.8 | 64.5 | 36.8 | 3.1 | 1599.1 | 1687.7 |

**Table 8.** Quantitative results on the Paris dataset with a magnification factor of 4.

| Indexes | Bicubic | GLP | HySure | PALM | CNMF | SDSR | SRMDNF | Proposed |
|---|---|---|---|---|---|---|---|---|
| MRMSE | 19.19 | 10.87 | 18.31 | 12.08 | 9.81 | 14.84 | 16.91 | **9.36** |
| MPSNR | 14.62 | 24.03 | 20.10 | 18.91 | 24.86 | 21.61 | 18.79 | **26.03** |
| MSSIM | 0.3060 | 0.8424 | 0.6053 | 0.8174 | 0.8727 | 0.7444 | 0.4091 | **0.8796** |
| ERGAS | 7.35 | 4.20 | 7.56 | 4.78 | 3.89 | 5.79 | 6.52 | **3.64** |
| UIQI | 0.2681 | 0.8112 | 0.5461 | 0.7334 | 0.8434 | 0.6500 | 0.4928 | **0.8648** |
| SAM | 0.1257 | **0.0745** | 0.1817 | 0.0890 | 0.0904 | 0.1821 | 0.1136 | 0.0958 |
| Time | 0.3 | 3.7 | 8.7 | 20.7 | 2.9 | 0.7 | 527.1 | 539.5 |

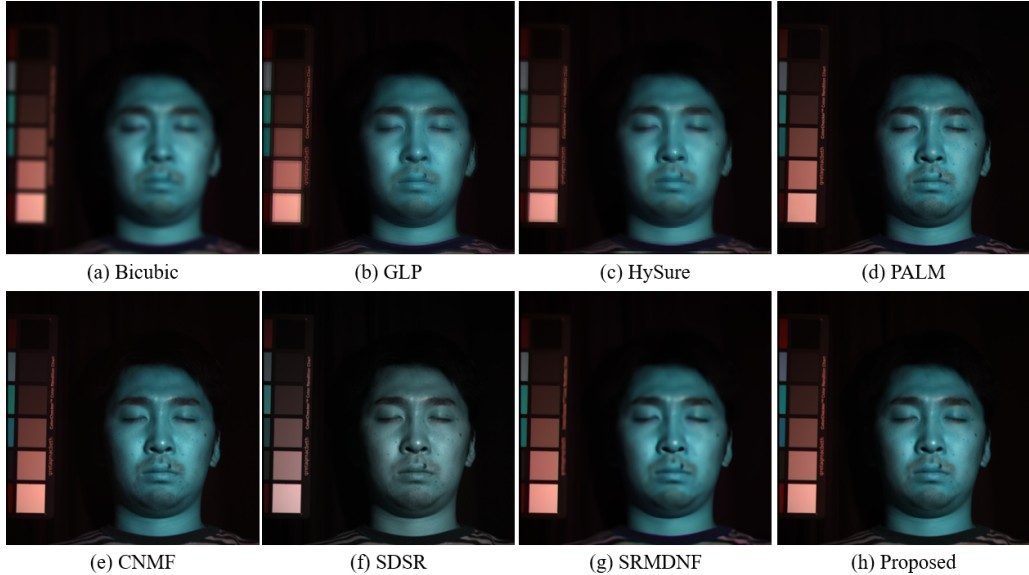

**Figure 10.** Super-resolution reconstruction results on the Face image from the CAVE dataset with a magnification factor of four. (**a**) Bicubic (MSSIM: 0.9205); (**b**) GLP (MSSIM: 0.9691); (**c**) HySure (MSSIM: 0.9427); (**d**) PALM (MSSIM: 0.9693); (**e**) CNMF (MSSIM: 0.9348); (**f**) SDSR (MSSIM: 0.9561); (**g**) SRMDNF (MSSIM: 0.9666); (**h**) proposed (MSSIM: **0.9782**).

## 5. Conclusions

In this paper, a novel domain transfer learning-based HSI super-resolution method was proposed to improve the spatial-resolution of the observed HSI, on the basis of the spatial similarity and spectral consistency. First, the proposed method obtains the transferred high-spatial-resolution HSI by exploiting the knowledge from the natural images. Then, the transferred high-spatial-resolution HSI, the observed low-spatial-resolution HSI, and high-spatial-resolution MSI are unified to estimate the endmember and abundance matrices during the spectral mixture analysis. Finally, the optimal endmember and abundance matrices are used to construct the target high-spatial-resolution HSI.

Through the experiments on the three real-world HSI datasets, i.e., CAVE, Pavia, and Paris datasets, the proposed method achieved better super-resolution reconstruction performance than the competing approaches. Specifically, the performance of the proposed method can still outperform the baselines for the magnification factors of three and four, which indicates that the proposed method is suitable for complex real-world super-resolution applications. When compared with the bicubic interpolation with a magnification factor of three, the average MSSIM values obtained by the proposed method at least increased by 4.4%, 40.1%, and 55.5% for the CAVE, Pavia, and Paris datasets, respectively. The proposed method captures the nonlinear mapping relationship between low- and high-resolution natural images and then transfers the knowledge to the HSI domain. In addition, the proposed method enhances the spatial similarity and preserves the spectral consistency in the optimization of the endmember and abundance matrices. Therefore, the proposed method achieved qualitative and quantitative results in the experiments.

In the future, the focus of our work will be on how to estimate the endmember matrix or the abundance matrix with higher precision under some effective constraints.

**Author Contributions:** All the authors designed the study and discussed the basic structure of the manuscript. X.L. carried out the experiments and finished the first version of this manuscript. L.Z. provided the framework of the experiments. J.Y. reviewed and edited this paper.

**Funding:** This work was supported in part by the National Natural Science Foundation of China under Grants 61601416, 61771349, U1536204, and 41701417, the Fundamental Research Funds for the Central Universities, China University of Geosciences, Wuhan, under Grant CUG170612, the Hong Kong Scholars Program under Grant XJ2017030, and the Hong Kong Polytheistic University Research Fund.

**Conflicts of Interest:** The authors declare no conflict of interest.

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
