# Peer review of "Domain Transfer Learning for Hyperspectral Image Super-Resolution"

_remotesensing, doi:10.3390/rs11060694_

Round 1

Reviewer 1 Report

The paper is fine, the maths describing the methods are sound. Results are slightly better (in general) than compared methods. This may justify publication.

My opinion, anyway, concerns the true novelty of the paper that seems (to me) insufficient, as the proposed approach mainly regards the application of transfer learning of a Deep Neural Network to this specific problem. Maybe a letter instead of an article might be the best positioning for this submission. 

Reviewer 2 Report

This paper presented a new framework based on domain transfer learning and exploited the cross-correlation between the observed low-resolution HSI and high-resolution MSI, and achieves comparable performance on estimated HSI. It is quite useful for HIS improvement because it provides another possibility for low resolution image processing. It is uniquely structured that the workflow contained both deep learning and image resolution processing.The contents in every paragraph and comparison process seem to be able to generate persuasive instructions. The setting of the research process and the selection of research indicators are also in line with academic requirements. However, some calculus processes should be wrote in clear way to make sure the research can be verified by others. There should be more discussion and explanation on whether the research processed have broad applicability. Basically, the professionalism and creativity of this article is worthy of recognition. It would be a good article for publishing. Here with concerns need to be addressed:

Question & Comment 1:

The main core function of CNN is feature comparison and extraction. However, the deep learning technology has some black box testing or gray box testing characteristics in nature, so it may block some meaningful information during the calculation process. It will be worth discussing in more detail about the uncertainty.

Question & Comment2

In the process of CNN calculus, a small-scale result graphics are often produced due to convolution sampling process, and some boundary characteristics might be lost. How can you prevent or solve this problem?

Question & Comment 3:

At present, some multi-spectral satellite images usually have comparable resolutions with aerial photos or orthogonal images, but are relatively easy to generate more noise such as salt and pepper effect. Is it possible to get the poor estimated HSI because noises are amplified in abundance matrix?

Question & Comment 4:

This article did not seem to discuss and compare the calculation time and performance of various algorithms. It may take more than one day to finish the calculation with CNN algorithm mentioned in the article the based on hardware specifications example. The eesearch results have been able to produce better estimated HSI, but how to increase the feasibility of application in the future is worth discussing.

Question & Comment 5:

Did the complexity of the image contents and the complexity of the feature arrangement affect the research results?

Reviewer 3 Report

1- Page 2, Line 34, Check this (the HSI data can formulated by)! grammatical error.
2- Page 2, Lines 34-39, rewrite this paragraph.
3- Confusion between HSI and MSI, please recheck it. (title of Fig 1, line 43, line 64).
4- I think you have to increase the number of related works, and you have to talk about these related works (give more details about what they have presented). and I suggest you to use some of these articles:
i- Super-Resolution of Hyperspectral Image via Super pixel-Based Sparse Representation, Neurocomputing (2017). (explain what are the differences of yor proposed method with respect of this one).
ii- Hyperspectral Image Super-Resolution Based on Spatial and Spectral Correlation Fusion," in IEEE Transactions on Geoscience and Remote Sensing, vol. 56, no. 7, pp. 4165-4177, July 2018. (explain what are the differences of yor proposed method with respect of this one).
5- Page 7, Line 210, give the reasons for blurring the original HSI and then down sampling the result with a magnification factor r ?
6- There are some typos, for example, Line 252 (Check all the manuscript, please)
7- Give some citations which have used these quantitative indexes (articles used these indexes to evaluate the quality of reconstructed images), and define which of these indexes are subjective and which of them are objective, if possible. For example:
i- Image modeling based on a 2-D stochastic subspace system identification algorithm.
ii- Enhancement of Curve-Fitting Image Compression Using Hyperbolic Function.
iii- Image Quality Assessment: From Error Visibility to Structural Similarity.
8- From Tables 1 and 2, give explanation for the increasing time of your proposed method (1990.3) for Pavia dataset, while the time of all other methods have been decreased !
9- I suggest you to put the results of average values of RMSEs in table form instead of writing it in the title of figures 6-9, what do you say?

Reviewer 4 Report

This manuscript describes a new approach for hyperspectral image super-resolution.  The first step uses neural nets trained on natural color photographs to generate an initial high resolution HSI image, which is then combinined with spectral unmixing technique in a three-term non-negative optimization.  This approach seems good, and I liked the way the data is combined in optimization problem of equation (9).  One would expect even better results using a neural net trained on imagery more relevant to the domain, but this paper shows the flexibility of this approach.

I have a few general comments.  I thought it was odd to take a neural net trained on natural color images and apply it a single band of the HSI (replicated three times to form grey scale image).  It seems this is wasting some color information. Were any other options tried?

Are there any requirements on overlap of HSI and MSI in the spectral domain?  That is, does the HSI have to overlap and span include all of the bands of the MSI?

My largest criticism of the paper is in the experimental methodology.  The optimal values of alpha and beta parameters are obtained for each data set using the actual ground truth.  In practice, this would not be possible.  One must set the values of alpha and beta without the benefit of having the ground truth to determine the optimal alpha and beta.  In this respect, the experimental results may be inflated over what one could expect for a new image (previously unseen).

The writing in the manuscript is good; I found only a couple of typos or problems in the text:

1) Line 41: "Since the observed HSI and the target HSI capture the same scene, their endmember matrices should be the same."  This is not guaranteed if there is a large time delay between the acquisition of the two images (e.g., due to seasonality, weather, clouds, urban changes, etc.).  It would be good to state some limits related to this.

2) Line 57: Define natural images in the Introduction (also in the abstract) because it is an overloaded term.  It could mean natural color images (i.e., RGB) of aerial or satellite imagery, but this is not what is meant.

3) Line 252: Typo: "third"

4) Line 309: "noise" instead of "noises"

Reviewer 5 Report

Thank you to the Authors for having improved the quality and content of their original manuscript.

I believe that a few minor corrections could help improving further the quality of this revised manuscript. They are reported just below:

1)  Natural images are often 8-bit images while remotely sensed images are rather 12-bit or 16-bit images. This means that both the range and the distribution of pixel values within the corresponding range may differ significantly for these two kinds of images. This should be discussed here as knowledge learned from natural images is transferred for enhancing the ability of HSI super-resolution.

2) Specify the coordinates of the region of interest of size 200x200x3 selected from the Pavia image

3) Specify systematically the wavelength of channels chosen for pseudo color view (where this remark applies)

4) Add standard deviation(s) to average values reported in Table 5

A few typo(s) detected:

Hence, enhacing the spatial resoultion

ss following equation (8)

In addition, the value of HSI data if often large,

For a pair comparsion, the number of endmembers C in this paper is set as 10

Round 2

Reviewer 2 Report

Overall, the idea flow and results of the manuscript is clearer and can be more followed. 

All the comments had reply with acceptable correction.

Reviewer 3 Report

I'd like to present my thanks to you because you have followed all my previous comments.

Author Response

Thank you very much for your comments and suggestions.

Reviewer 4 Report

The revised manuscript is improved, and my questions were answered.  Thank you.

Author Response

Thank you very much for your valuable comments and suggestions.

Reviewer 5 Report

The Authors have correctly replied to almost all my comments and suggestions.

One typo remains: For a pair comparison, -> For a fair comparison

Both the quality and content of this revised manuscript have been improved substantially.

I do not have further questions or remarks.